# Content- and Topology-Aware Representation Learning for Scientific Multi-Literature

**Kai Zhang[1,2], Kaisong Song[2,3], Yangyang Kang[2*], Xiaozhong Liu[1*]**
[1]Worcester Polytechnic Institute, Worcester, USA
[2]Alibaba Group, Hangzhou, China
[3] Northeastern University, Shenyang, China
{kzhang8, xliu14}@wpi.edu, {kaisong.sks, yangyang.kangyy}@alibaba-inc.com

## Abstract

Representation learning forms an essential building block in the development of natural language processing architectures. To date, mainstream approaches focus on learning textual information at the sentence- or document-level, unfortunately, overlooking the inter-document connections. This omission decreases the potency of downstream applications, particularly in multi-document settings. To address this issue, embeddings equipped with latent semantic and rich relatedness information are needed. In this paper, we propose SMRC$^2$, which extends representation learning to the multi-document level. Our model jointly learns latent semantic information from content and rich relatedness information from topological networks. Unlike previous studies, our work takes multi-document as input and integrates both semantic and relatedness information using a shared space via language model and graph structure. Our extensive experiments confirm the superiority and effectiveness of our approach. To encourage further research in scientific multi-literature representation learning, we will release our code and a new dataset from the biomedical domain[1].

## 1 Introduction

With the increasing amount of scientific publications, researchers progressively turn to intricate natural language processing (NLP) tools for paper discovery and recommendation. However, most existing tools are tailored towards specific downstream tasks(Lu et al., 2020; Viswanathan et al., 2021) and thus their capabilities seem limited when presented with more complex tasks. This is attributed to the focus on either the content or topology information of scholarly documents, instead of both.

An aspiring scholarly search engine offers researchers with the capacity to comprehend query topics and access multiple appropriate publications, or even more innovatively, a synthesized summary of the overall set of sources. Such capabilities have the potential to revolutionize the modern research process (Wang et al., 2019; Zhang and Liu, 2016). Unfortunately, existing methods only focus on single document representation learning (SDRL) and the recommended paper are selected based on the rank of similarity between the learned representation and the embedding of input query.While language models, such as Bert family(Devlin et al., 2018; Maheshwari et al., 2021; Lee et al., 2020) and GPT family(Brown et al., 2020; Ouyang et al., 2022), has seen tremendous progress in understanding scientific documents (Wei et al., 2022) and graph-based approaches like GraphFormer(Yang et al., 2021), SPECTER(Cohan et al., 2020) have studied the relatedness of a scientific paper in its citation networks, little effort has been devoted to exploring scientific literature representation learning at the multi-document level (MDRL). In this paper, we extend the community's attention of representation learning to multi-document level and explore how to learn both semantic and relatedness information from content and topological networks.

Intuitively, knowledge entities and text content of the given paper can provide latent semantic information. For example, we could simply infer a paper belongs the biomedical domain by knowing its entities such as "*diabetes*", "*GPX1*" (a gene), etc. However, there is a risk of incorrect inference when a computer science paper takes biomedical instances as examples. Same situation for using text content only, one can classify a paper simply by reading its abstract. Unfortunately, an NLP model will be hallucinated due to the close lexical similarity of papers from different categories. Thus, how to jointly learn from both entities and paper content can be critical to provide latent semantic information. Typically, entity-relation triplets will be modeled in the form of a graph which poses a challenge

---

*Corresponding Author
[1]https://github.com/MatthewKKai/SMRC2

of balancing the learning between graph neural networks (GNNs) and language models (LMs). We novelly introduce Wasserstein distance to enable GNN and LM in parallel and interactively learn from each other while ensuring the learning of the latent semantic information.

By leveraging the multi-document relations, e.g., citation graph, graph neural networks can simply enable each scientific paper to learn from its neighbors via the message passing mechanism. For example, An et al. (2021) proposed a citation graph-based memorization model (CGSUM) to incorporate the information of both the source paper and its references to obtain summarization. Wang et al. (2022b) utilized a disentangled representation based model DisenCite to encode paper according to both textual contexts and structure information for citation recommendation and generation. These works share resemblances to our method since they also take multi-document as input. However, our work aims to explore output representation equipped with latent semantic and rich relatedness information for each document, as well as a unified embedding for the entire input. This approach enables the learned representations not to be limited to specific tasks. Inspired by the effective masking strategy(Hou et al., 2022), we propose to use a graph neural network equipped with a novel masking strategy to enable each document to learn the relatedness of its neighbors.

To this end, we propose SMRC$^2$ which learns the **S**cientific **M**ulti-literature **R**epresentation from both **C**ontent information under Wasserstein constraint and **C**itation information via graph structure. The main contributions of our work are a threefold:

• ***Conceptual:*** We extend scientific representation learning to multi-doc level and demonstrate its necessity in various downstream tasks as well as the main difference compared with existing work.

• ***Methodological:*** The proposed SMRC$^2$, incorporates a Wasserstein constraint to optimize the multi-view content learning and a graph masking strategy to complete the multi-document topology learning which enable us to learn both latent semantic and rich relatedness information from a scientific ecosystem.

• ***Experimental & Resource:*** We conduct extensive experiments using three benchmarks on three tasks and outperform several state-of-the-art baselines. We also release a new *Bio-Sci* dataset that can be used for multiple downstream tasks.

## 2   Related Work

**Representation Learning** Most existing representation learning works in NLP focus on capturing the dependency in word-level, token-level and sentence-level(Peters et al., 2018; Devlin et al., 2018; Wu et al., 2020). With the success in previous work, some efforts have been made to extend the learning paradigm to document-level(Yasunaga et al., 2022a; Li et al., 2021). However, these methods suffer from the massive noises introduced by irrelevant words/sentences. Several Context-based and Graph-based models are proposed(Jeong et al., 2020; Wang et al., 2022b) to reduce noises in the training stage. Given a document and its salient entities, for example, graph-based models can easily build a graph where the entities are nodes and their relations are edges to derive/infer the knowledge within this graph. Unfortunately, these methods lack the consideration of rich relatedness in citation networks. Recent works (Cohan et al., 2020; Yasunaga et al., 2022b; Yang et al., 2021) push the boundary forward by including citation information. These works are orthogonal to ours as the input to our model is also multi-literature with citation relations, but the difference is that they leverage citation information for enhancing the source document's representation while our work is forcing each document to learn both semantic and relatedness information and at the same time, output a unified representation for the entirety. Further, research work(Zhao et al., 2022; Chien et al., 2021; Duan et al., 2022; Sun et al., 2020; Kong et al., 2022) proposed to tackle the Text-Attributed Graph learning task (TAG) are also similar to our approach which aims at better schema for jointly graph and language model learning. However, our work goes beyond these approaches by explicitly integrating semantic and relatedness information in scientific multi-literature representation.

**Multi-Document as Input** In real-world scientific document applications, obtaining information from multiple documents is a common requirement. Researchers may want to know the whole story of a topic without reading every paper related to it, which falls under the subfield of NLP known as multi-document summarization (MDS). While other multi-document tasks, such as multi-document reading comprehension, which aims to understand the content of multiple documents, are also of interest, our focus in this paper is on MDS to evaluate the learned multi-document rep-

resentation. Extractive methods usually produce a summary by selecting the ranked sentences from the given document set(Wan et al., 2015; Mendes et al., 2019; Zhong et al., 2020). For example, Nallapati et al. (2017) treats MDS as a sequence classification problem where each sentence is visited in sequential order and then adopts an RNN-based model to decide whether the sentence should be included in the summarization or not. Liu et al. (2019) conceptualizes MDS to induce a multi-root dependent tree representation of the documents. However, similar sentences may be close in the vector space and share close scores which would cause redundancy problem(Narayan et al., 2018). Recently, graph-based approaches which aim to extract salient textual units from documents based on graph structure representations of sentences are proposed to eliminate this(Chen et al., 2021; Pasunuru et al., 2021). Different from these works, our method leverages document representation with latent semantic information as nodes and their citation relations as edges to construct the multi-document level graph. Then, the message passing mechanism can be resorted to equipping relatedness information for the final representation which can be applied in multi-document tasks.

## 3  Methodology

**Problem Formulation** Before going further, we first give preliminary definitions of concepts in this paper. Let $\mathcal{P} = \{P_1, ......, P_N\}$ denotes a paper group with $N$ papers and $\mathcal{G}_{\mathcal{P}} = (\mathcal{V}, \mathcal{A}, \mathcal{X})$ is the multi-doc graph of $\mathcal{P}$ with each node $v_i \in \mathcal{V}$ represents a paper, $\mathcal{A} \in \{0, 1\}^{N \times N}$ is the adjacency matrix and $\mathcal{X} \in \mathbb{R}^{N \times d}$ is the node feature matrix. Moreover, given $f_{GE}$ as the graph encoder, $f_{LM}$ as the language model, and $f_{GD}$ as the graph decoder, our goal here is to obtain the document representation $H_i$ as well as the multi-document representation $H_M$ as follow:

$$
\begin{aligned}
h_i &= [f_{GE}(\mathcal{P}), f_{LM}(\mathcal{P})] \\
H_M &= f_{GD}(\mathcal{G}_{\mathcal{P}}(H))
\end{aligned}
\tag{1}
$$

where $h_i \in H$, $[\cdot, \cdot]$ is the concatenation operation.
**Overview** As can be seen in Figure 1, SMRC$^2$ consists of two modules: Semantic representation learning module and Relatedness representation learning module. In the semantic learning module, SMRC$^2$ first in parallel obtain the entity-relation

graph embedding via a GNN encoder and the abstract text[2] embedding via a language model. These two levels of information are then concatenated to form the initialized node features, along with their citation relations, to create a multi-doc graph. To learn rich relatedness information, we randomly mask some of their citation relations and train a GNN-based decoder to recover them based on the node features using the message passing mechanism. By leveraging both the latent semantic and rich relatedness information, SMRC$^2$ is capable of not only single document but also multi-document applications.

### 3.1  Semantic Information Learning

Intuitively, the content of a scientific paper contains abundant semantic information. However, using the entire text content can be challenging since it may exceed length and computation resource limits, and would also introduce significant noise. Various attempts have been made to learn semantic representation from different aspects such as entities(Pivovarova and Yangarber, 2018) and abstract(Kim and Gil, 2019) etc. Our hypothesis is that entities can provide a knowledge backbone and abstract can provide an overall summary which should be modeled and learned interactively together. Hence, we utilize two encoders for both entities and text, and novelly introduce the Wasserstein constraint to jointly train them.

**Entity-Level** Salient entity information can be used to somehow infer the content of a given paper. For instance, consider a scientific paper such as Zeng et al. (2020). If you know the "Dataset (e.g.,**DocRed**)", "Metrics (e.g., **F1 score**)", "Task (e.g., **Relation Extraction**)", "Method (e.g., **GAIN**)", you can conclude that this paper is using GAIN to tackle relation extraction task and evaluate their approach on DocRed with F1 scores. Motivated by this, we follow Neumann et al. (2019); Jain et al. (2020)'s work and extract the entities within the given paper. Then we retrieve the relations among entities from a pre-defined relation set and use the extracted entities as nodes and their relations as edges to build an entity-level graph. Then we apply a two-layer GCN to obtain the representation $h_i^E$ as shown in Eq.2.

$$
h_i^{E(l)} = ReLU((\mathcal{A} + I)h_i^{E(l-1)}W_E^l)
\tag{2}
$$

---

[2]Other sections are also considered, however, experiments show that they would introduce noises and require more computational resources.

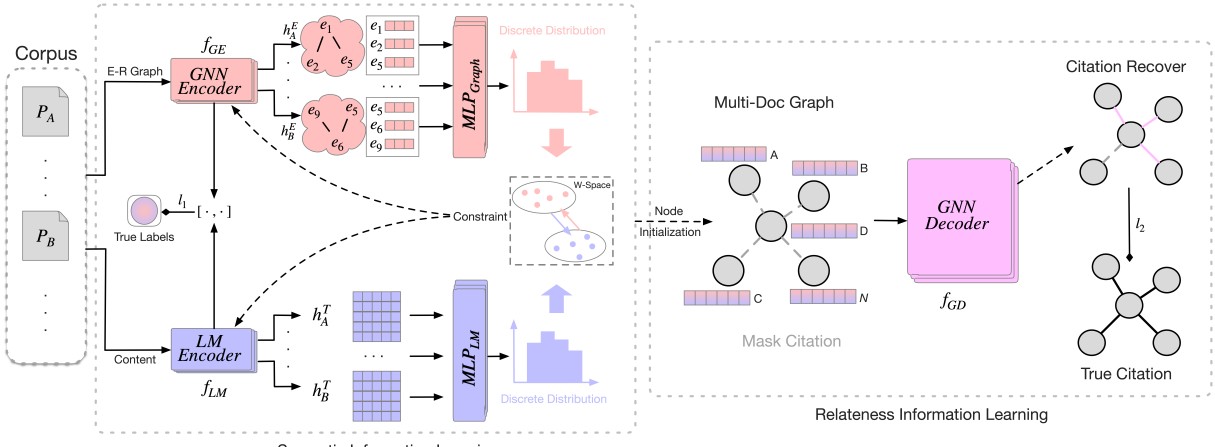

Figure 1: The overall frame for SMRC$^2$ involves first obtaining semantic embeddings for each document in a paper group via the semantic information learning module. Then these embeddings serve as the initialized node features and their citation relations serve as the edges to form the multi-doc graph, which can later learns the relatedness information. The concatenation operation is denoted as $[\cdot, \cdot]$.

where $h_i^{E(l)}$ is the l-th layer's hidden embedding ($h_i^E = h_i^{E(l)}$, $h_i^{E(l)} = [h_i^{e_1(l)}, ..., h_i^{e_m(l)}]$ where $h_i^{e_m(l)} \in \mathbb{R}^{n \times d}$ and $m$ is the number of entities), I is the identity matrix and $W_E^l$ is trainable parameters. Notably, for entity starting from s-th word to t-th word, $h_i^{e_m(0)} = \frac{1}{t-s+1} \sum_{j=s}^{t} g_j$, where $g_j$ is the word embedding.

**Text-Level** However, relying solely on entity-level information may not be sufficient for representing the entire paper, as prescribed in Section 1. Therefore, we opt to incorporate the abstract text since it contains the most general content of a scientific paper. Given the abstract of a scientific publication $P_i$, a language model will generate a d-dimensional embedding vector denoted as $h_i^T \in \mathbb{R}^d$:

$$h_i^T = f_{LM}(P_i^{abs}) \qquad (3)$$

where $f_{LM}$ is a language model.

**Entity-Text Interaction** Given $\mathcal{P}$, the corresponding distribution of document labels $\mu(\mathbf{y}) = \mu(\{y_1, y_2, ..., y_N\})$ can be the supervised signals for $f_{GE}$ and $f_{LM}$ to learn salient semantic information. However, relying solely on abstract or entity information for predicting labels can be unreliable. To overcome this limitation, we introduce a novel approach using Wasserstein Distance (WD) to restrict $f_{GE}$ and $f_{LM}$ from learning independently of each other while ensuring they predict accurate labels for the documents. For LM and GNN, the predicted distribution can be described as follows:

$$\mu^T(\mathbf{y}^T|\mathcal{P}) = Cat(y_i^T|\phi(MLP(h_i^T)))$$
$$\mu^E(\mathbf{y}^E|\mathcal{P}) = Cat(y_i^E|\phi(MLP(h_i^E))) \qquad (4)$$

where $\phi$ is a Softmax function, MLP is a multi-layer perceptron.

Kullback-Leibler divergence(Joyce, 2011) and other likelihood-based divergence methods suffer from statistical limitations such as invariant to any invertible transformation(Ozair et al., 2019). They are also not suitable for representation learning due to their sensitivity to the trivial differences in the data samples. However, inspired by the success of WD in GAN training(Arjovsky et al., 2017), we consider using WD to enable $f_{GE}$ and $f_{LM}$ to learn from each other by minimizing the distance between their predicted label distributions. Wasserstein Distance is a metric-aware divergence derived from transport optimal theory. It measures the discrepancy between two distributions in terms of minimum total costs associated with some transport function(Hou et al., 2020). Drawing inspiration from Villani et al. (2009), we give the definition of Wasserstein Constraint in our settings:

**Definition 1.** *Given two distributions $\mu_1$, $\mu_2$, a cost function cst: $\mathcal{N} \times \mathcal{N} \to \mathbb{R}$ and the set of joint probability $\eta(\mu_1, \mu_2)$, the **Wasserstein Distance** can be defined as:*

$$W_{cst}(\mu_1, \mu_2) = \inf_{\gamma \in \eta(\mu_1, \mu_2)} \int_{\mathcal{N} \times \mathcal{N}} cst(y_1, y_2) d\gamma \quad (5)$$

where $y_1$ and $y_2$ are data points in distributions $\mu_1$ and $\mu_2$, respectively.

Therefore, our current objective can be reformulated as minimizing the optimal cost function while simultaneously forcing the encoders to learn semantic information for accurate document label

prediction. Assuming that the optimal cost function can be replaced with a certain metric $d$ or its $k$-th power $d^k$ with $k \geq 1$, then the Wasserstein Loss of our settings can be defined as:

$$W_d^k(\mu^T, \mu^E) = \inf_{\gamma \in \eta(\mu^T, \mu^E)} \mathbb{E}(d^k(y_i^T, y_i^E), \gamma) \quad (6)$$

where the set of joint distribution is $\eta(\mu^T, \mu^E) = \{\gamma \in \mathbb{R}_+^{\mathcal{N} \times \mathcal{N}} : \gamma \mathbf{1} = \mu^T, \gamma^T \mathbf{1} = \mu^E\}$, $\mathbf{1}$ is the all-one vector. By calculating the derivative of $\mu^T$, $\mu^E$ with respect to $W_d^k$ in an attempt to approach true document label distribution, we can describe the overall loss function of semantic representation learning module as:

$$\mu(\hat{\mathbf{y}}) = Cat(\hat{y}_i | \phi(MLP([h_i^T, h_i^E])))$$
$$l_1 = W_d^k + (-\sum_{i \in N} P(y_i) log P(\hat{y}_i)) \quad (7)$$

where the second term is the Cross-Entropy loss. By incorporating the Wasserstein Constraint, the two encoders can enhance their learning ability by leveraging each other's outputs, while also capturing the necessary semantic information for document classification. As far as our knowledge extends, this is the first attempt to introduce Wasserstein Distance for multi-view semantic representation learning within a scientific document.

### 3.2 Relatedness Information Learning

To achieve effective multi-document representation learning, it is imperative to consider the rich relatedness information of the topological network. While the knowledge of each paper's semantic information can aid in determining its content, it may be challenging to produce summaries based solely on this information. Relying solely on semantic representation may also lead to sub-optimal results, especially in cases with insufficient textual features. Therefore, it is crucial to capture the semantic information while also incorporating the rich relatedness information. A promising aspect of Graph Neural Networks (GNNs) is their ability to be aware of their neighbors' facts through the message-passing mechanism (details can be seen in Appendix 5), which can be described as follows:

$$h_i^{(l)} = \sigma(AGG(MSG(h_{NB(i)}^{(l-1)}), \mathcal{A})) \quad (8)$$

where $h_i^{(0)} = [h_i^E, h_i^T]$, $\sigma$ is an activation function (e.g., ReLU), $MSG(\cdot)$ and $AGG(\cdot)$ stand for the message and aggregation functions respectively,

NB(i) is the neighbor nodes of i. We can create $\mathcal{G}_\mathcal{P}$ by considering each document in $\mathcal{P}$ as a node and their citation relations as edges. Then the message-passing mechanism can be leveraged to enable each document to learn about the relatedness within its citation network. Intuitively, one can judge the possibility of citation relation between two papers by knowing their semantic content (e.g., if two papers are all about using graph to learn document representation, there exists a chance that they have citation relation). Inspired by the success of masking strategy on Variational Graph Auto Encoder (VGAE) to predict the adjacency matrix(Hou et al., 2022), we employ a similar approach. We intuitively mask some of the citation relations and train a VGAE to recover the citation relations. However, the difference is that we don't need the encoder part of VGAE and instead of an unsupervised process, we adopt self-supervised settings. Specifically, we first initialized a random masking matrix $\mathcal{M}$ with masking ratio $\omega$ and obtain the masked adjacency matrix $\mathcal{A}_{masked}$ by applying element-wise multiplication between $\mathcal{M}$ and $\mathbf{A}$ (e.g., $\omega$ of $\mathcal{M}$ are zeros and the reset are ones). Then given the semantic representation of each document as the initial node embedding, GNN can iteratively update them by applying Eq. 8 on $\mathcal{G}_\mathcal{P} = (\mathcal{V}, \mathcal{A}_{masked}, [h_i^T, h_i^E])$. Now, our goal here becomes reconstructing the $\hat{\mathcal{A}}$ in terms of learned $l$-th node embedding. However, directly applying the final-layer hidden representation $h_i^{(l)}$ can cause performance degradation for producing $\hat{\mathcal{A}}$ since there would be long-distance information lost and irrelevant inclusion problems as the number of layers increase, hence we aim to capture the cross-correlations for every node-pair:

$$h_{ij} = [h_i^0 \odot h_j^0, ..., h_i^l \odot h_j^l] \quad (9)$$

where $\odot$ denotes pair-wise element multiplication, $i, j \in [0, N]$. We now can reconstruct $\hat{\mathcal{A}}$ as:

$$p(\hat{\mathcal{A}} | \mathbf{h}_{ij}) = \prod_{i=1}^{N} \prod_{j=1}^{N} p(\hat{\mathcal{A}}_{ij} | h_{ij})$$
$$with \ p(\hat{\mathcal{A}}_{ij} = 1 | h_{ij}) = \sigma(h_{ij}^T h_{ij}) \quad (10)$$

where $\hat{\mathcal{A}}_{ij}$ is the element of $\hat{\mathcal{A}}$ and $\sigma$ is an activation function. By having $\hat{\mathcal{A}}$ and $\mathcal{A}$, we follow Kipf and Welling (2016b)'s work to optimize $f_{GD}$ as:

$$l_2 = -\sum_{i \in \mathcal{N}} p(\mathcal{A}) log \ p(\hat{\mathcal{A}}) \quad (11)$$

| Task | Type | Dataset | | | |
|---|---|---|---|---|---|
| | | Ogbn-Arxiv | Ogbl-Citation2 | Multi-XScience | Bio-Sci |
| Document Classification | SDT | ✓ | - | - | ✓ |
| Citation Prediction | | - | ✓ | - | ✓ |
| Multi-doc Summarization | MDT | - | - | ✓ | ✓ |

Table 1: Different tasks on different datasets

| Model | Type | Ogbn-Arxiv | Bio-Sci |
|---|---|---|---|
| Bio-Bert | LM | $68.21 \pm 0.17$ | $75.67 \pm 0.17$ |
| Sci-Bert | | $73.80 \pm 0.12$ | $70.49 \pm 0.31$ |
| EnGCN | Graph | $77.57 \pm 0.07$ | $85.27 \pm 0.06$ |
| GraphSAGE | | $71.49 \pm 0.21$ | $79.35 \pm 0.07$ |
| GIANT-XRT | LM+Graph | $76.37 \pm 0.11$ | $84.81 \pm 0.13$ |
| GLEM(Graph) | | $75.50 \pm 0.24$ | $82.93 \pm 0.11$ |
| GLEM(LM) | | $74.53 \pm 0.12$ | $78.91 \pm 0.07$ |
| **SMRC$^2$** | LM+Graph | $\mathbf{77.66 \pm 0.17}$ | $\mathbf{88.41 \pm 0.21}$ |

Table 2: Mean accuracy (%) $\pm$ one standard deviation comparison for Document Classification in terms of different types of baselines.

Now, with the latent semantic and rich relatedness information, the learned representation is $h_i = h_i^{(l)}$ according to Eq. 8 and $H_M = [..., h_i, ...]$.

### 3.3 Training

Overall, we consider the semantic representation learning module as a supervised process and the relatedness representation learning module as a self-supervised process, the whole pipeline of SMRC$^2$ can be trained through $l$ as follows:

$$l = l_1 + l_2 \qquad (12)$$

We perform full-batch gradient descent on $l$ which enables SMRC$^2$ to obtain both semantic and relatedness information within supervised plus self-supervised learning settings for scientific multi-literature representation learning.

## 4 Experiments

In this section, we empirically evaluate the performance of our learned document representation $h_i$ and the unified multi-doc representation $H_M$.

### 4.1 Tasks & Datasets

**Tasks** The key distinguishing aspect of our work from previous research is that our learned representation is applicable not only to single-document tasks (SDT) but also to multi-document tasks (MDT). Thus, we have evaluated our learned representation on both SDT and MDT for four datasets, as shown in the Table 1:

**Document Classification (Doc Cla.)** which aims to predict the class for each paper using semantic or structural information. Here, we resort

to predicting the labels by $\hat{y} = \phi(MLP(h_i))$. We have selected several SOTA baselines of different types to assess the efficacy of our model: LM - Bio-Bert(Lee et al., 2020), Sci-Bert(Maheshwari et al., 2021); Graph - EnGCN(Duan et al., 2022), Graph-Sage(Kong et al., 2022); LM+Graph - GIANT-XRT(Chien et al., 2021), GLEM(Zhao et al., 2022).

**Citation Prediction (Cit Pre.)** which refers to predicting the citation relations by knowing the semantic content of each paper. Our model predicts citation relations by reconstructing the adjacency matrix $\mathcal{A}$ via Eq. 10. For evaluating the performance of our proposed model, we compare it against several SOTA baselines: GCN(Kipf and Welling, 2016a), GraphSage(Kong et al., 2022), GraphSaint(Zeng et al., 2019).

| Model | Ogbn-Citation2 | Bio-Sci |
|---|---|---|
| Full-batch GCN | $83.14 \pm 0.21$ | $71.41 \pm 0.17$ |
| GraphSage | $82.60 \pm 0.36$ | $69.17 \pm 0.31$ |
| GraphSaint | $79.85 \pm 0.39$ | $70.31 \pm 0.26$ |
| **SMRC$^2$** | $\mathbf{83.31 \pm 0.27}$ | $\mathbf{73.97 \pm 0.19}$ |

Table 3: MRR score in percent comparison for Citation Prediction in terms of different baselines.

**Multi-Document Summarization (MDS)** aims to generate a summary for given multiple documents. Here, we gauge the similarity between $H_M$ and the sentences from a sentence set derived from original papers. The top $k$ sentences will be picked and copied as the generated summary. SMRC$^2$ outperforms several baselines: HiMAP(Fabbri et al., 2019), Pointer-Generator(See et al., 2017), Hier-Summ(Liu and Lapata, 2019), KGSum(Wang et al., 2022a), REFLECT(Song et al., 2022).

**Datasets** We choose three benchmark datasets according to the tasks and would like to introduce a new benchmark dataset for the community - *Bio-Sci* which consists of 32,330 publications, 222,652 citation relations derived from PubMed Central[3]. Specifically, we follow the work of Achakulvisut et al. (2020) to extract the abstract, introduction and citation sentence for each paper (Notably, we only use abstract for LM in this work). Further, every

---

[3]https://ftp.ncbi.nlm.nih.gov/pub/pmc/oa_bulk/oa_comm/

| Model | Type | Multi-XScience | | | Bio-Sci | | |
|-------|------|---------|---------|---------|---------|---------|---------|
| | | ROUGE-1 | ROUGE-2 | ROUGE-L | ROUGE-1 | ROUGE-2 | ROUGE-L |
| BioBertABS | | 24.77 | 4.01 | 21.67 | 31.66 | 5.91 | 28.43 |
| SciBertABS | Concat | 25.11 | 4.17 | 22.91 | 30.02 | 5.04 | 27.60 |
| REFLECT | | 29.41 | 6.11 | 24.91 | 34.17 | 6.67 | 29.57 |
| HIMAP | | 26.61 | 6.19 | 25.31 | 34.11 | 6.76 | 29.77 |
| HIERSUMM | Multi | 25.43 | 6.01 | 24.91 | 35.02 | 7.04 | 29.66 |
| KGSum | | 30.16 | 6.24 | 25.37 | 34.98 | 6.91 | 30.07 |
| **SMRC$^2$** | | **31.47** | **6.73** | **26.03** | **36.23** | **7.31** | **30.63** |

Table 4: Rouge-1, Rouge-2, Rouge-L score comparison with different baselines.

paper is associated with an id called pmid which enables us to find the citation relations by matching the pmid. For each paper, we use its MeSH term as its category label. Every paper group in *Bio-Sci* consists of 10 papers and their citation relations[4].

## 4.2 Implementation Details

For Bio-Sci, we follow the work of Wang et al. (2021) and use SciSpacy(Neumann et al., 2019) to obtain entity-relation graph can is detailed in Appendix 5, the base LM model is Bio-Bert(Lee et al., 2020). For other datasets, we follow the work of Ye et al. (2022) in the data processing stage and use Sci-Bert(Maheshwari et al., 2021) as the base language model. We leverage Adam(Kingma and Ba, 2014) as the optimizer. Hyperparameters setting can be seen in Appendix 5. For evaluation, we follow the commonly used metrics on these benchmarks. All experiments are conducted with PyTorch(Ketkar and Moolayil, 2021) and Transformers(Wolf et al., 2020) on 4 Tesla P100 GPUs.

## 4.3 Comparative Study[5]

As shown in Table 2, SMRC$^2$ has demonstrated a relatively 0.12% and 1.33% improvement over the best baseline on Ogbn-Arxiv and Bio-Sci, respectively. Generally, we can conclude that relying solely on either LM or Graph alone is not comparable in terms of results. This can be explained by the fact that plain text can introduce noise, while graph structure can mitigate this issue. Furthermore, the results highlight the effectiveness of jointly leveraging Graph and LM, which validate our efforts in modeling the interactions of different modules. However, EnGCN produced competitive results among all baselines which we attributed to its ensemble training schema from multi-view of input which boosts the convergence of the whole

model. Additionally, the citation prediction results demonstrate the importance of node features in inferring the connections among publications since our model outperforms other baselines by aggregating the semantic embedding first as the initial node features while other baselines use word embedding solely. Furthermore, the learned multi-literature embedding of our model holds great promise for multi-document tasks such as multi-document summarization due to its ability of aggregating both latent semantic and rich relatedness information into the final representation.

Table 4 confirms our assumption, as our model outperforms current SOTA baselines. Surprisingly, HiMAP, which adapts a pointer-generator model for weight computation over multi-document, and HierSumm, which leverages a ranking-based selection mechanism for sentence selection, perform better than pre-trained large language models. We believe this is because these two models focus on finding the most important parts over multi-document, whereas pretrained models inevitably suffer from noise by simply concatenating all documents together as a single document before performing summary generation. It's also very interesting that on Bio-Sci, the goal is to recover abstract, sentence selection based models such as HierSumm and SMRC$^2$ performs better than other models. We also observe that KGSum outperforms REFLECT which we attribute to the fact that REFLECT only focuses on sentence selection which ignores the fine-grained semantic information from entity level and also the rewriting process is finished based on the inner natural language understanding ability of chosen pre-trained model without awareness of topological information. However, it is evident that our method surpasses KGSum which we attribute to the fact that our method leverages a Wasserstein Constraint for modeling the interactions between entity-relation graph and semantic text information while KGSum learns only from salient

---

[4]Detailed comparison can be seen in Appendix 5

[5]All baseline, we use the best parameter settings reported in their paper and we keep the base language model the same for fair comparison

| Type | w/o | w/ | Doc Classification | Citation Prediction | Summarization |
|---|---|---|---|---|---|
| | LM | - | $78.91 \pm 0.14$ | $69.33 \pm 0.11$ | 19.11 |
| | Graph | - | $79.61 \pm 0.20$ | $67.47 \pm 0.10$ | 23.71 |
| SMRC$^2$(Part) | WD | - | $82.91 \pm 0.34$ | $69.71 \pm 0.02$ | 24.31 |
| | - | KL | $81.31 \pm 0.17$ | $71.27 \pm 0.14$ | 20.51 |
| | - | JS | $85.77 \pm 0.08$ | $72.41 \pm 0.22$ | 24.41 |
| SMRC$^2$(Full) | - | - | $\mathbf{88.41 \pm 0.21}$ | $\mathbf{73.97 \pm 0.19}$ | **26.03** |

Table 5: Ablation study on different components. w/o denotes without, w/ denotes with, all the metrics stay the same as in Comparative Study (For the Summarization task, the metric used is ROUGE-L).

### 4.4 Ablation Study

To further evaluate the completeness of SMRC$^2$, we conduct ablation study and the results are presented in Table 5. During experiments without LM, we use random embedding initialization to replace the initial embedding from LM which results in the huge drop of performance in Document Classification task compared to experiments without graph, showing the importance of semantic information. The citation prediction results reveal that solely using semantic information is not enough for inferring the connections while graph structure is necessary for including relatedness information. We can also witness the superiority of introducing Wasserstein Constraint as the performance drops when we apply KL or JS divergence as a replacement for semantic representation learning. Interestingly, we found that KL divergence is not stable during the training stage, with performance oscillation as the training iteration progresses, which we attribute to its asymmetry. Introducing Wasserstein Constraint not only solved the imbalance training problem but also enabled our model to learn semantic information from both text and entity aspects while forcing it to make correct label predictions, further confirming the necessity and superiority of the entire structure of our model.

### 4.5 Masking Study

| Settings | Doc Cla. | Cit Pre. | MDS |
|---|---|---|---|
| 10% | $86.19 \pm 0.11$ | $74.13 \pm 0.31$ | 24.91 |
| 20% | $\mathbf{88.41 \pm 0.21}$ | $\mathbf{73.97 \pm 0.19}$ | **26.03** |
| 30% | $84.17 \pm 0.07$ | $70.31 \pm 0.26$ | 22.51 |

Table 6: Masking study on evaluating the effectiveness of Masking Strategy, all the metrics stay the same as in Ablation Study.

We assess the effectiveness of our masking strategy applied in the relatedness learning module by setting different values of $\omega$. As can be seen in Table 6, we can conclude that masking ratio can influence the effectiveness of overall performance since the masking strategy is associated with the relatedness information learning. When masking more edges, the performance of citation prediction will inevitably decrease since the complexity of recovering can be higher. However, further experimentation revealed that 5% masking for citation prediction produced a result of $70.21 \pm 0.11$, suggesting a decrease due to the risk of overfitting. We can also notice that the performance on Document Classification and Multi-Doc Summarization tasks decreased as masking more edges since there can be semantic information lost and noise introduction due to the update of hidden representations which further validates the necessity of Eq. 9. Nevertheless, the recover process can be insufficient for the model to learn relatedness information when masking 10% of edges. The results suggest that masking 20% can be the best choice for SMRC$^2$.

### 4.6 Visualization

To further demonstrate the effectiveness of SMRC$^2$, we randomly select 100 papers for each category from *Bio-Sci* and utilize t-SNE(Van Der Maaten, 2014) to visualize these learned embeddings by SMRC$^2$ and compared them with the ones generated by BioBert. We set the perplexity and the number of iterations for t-SNE as 30 and 300, respectively. As shown in Figure 2, the clusters of papers from different categories generated by SMRC$^2$ are more compact than those by BioBert, indicating that our model is better at modeling semantic information. Our model's ability to learn rich relatedness information is also reflected by the inter-category distance. Intuitively, diseases that share high correlations should be close in the embedding space. For instance, the relationship between diabetes and kidney disease is accurately reflected, as

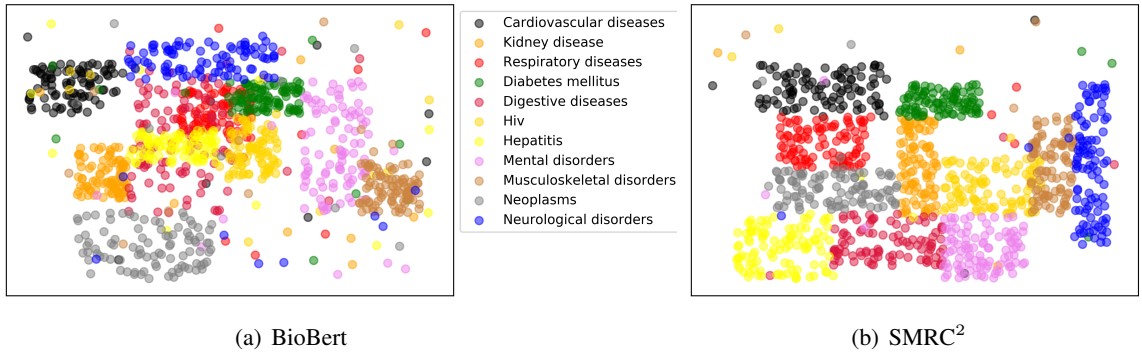

(a) BioBert

(b) SMRC$^2$

Figure 2: Representation visualization of selected papers using t-SNE.

the representations of "Diabetes mellitus(green)" and "Kidney disease(orange)" are close in the embedding space, which is consistent with the fact that diabetes is the leading cause of chronic kidney disease, and people with kidney disease are at a higher risk of developing diabetes. These findings further confirm the ability of SMRC$^2$ to capture latent semantic and rich relatedness information.

## 5 Conclusion

As a conclusion, we have advanced the field of representation learning to the multi-doc level for scientific literature, with the introduction of SMRC$^2$. This novel approach effectively combines two elements - semantic content and topological relatedness, to create a superior learning space. To address the imbalance learning problem that occurs during the semantic learning stage, we introduced Wasserstein distance. To our best knowledge, this is the first attempt to introduce WD for multi-view content learning. The extensive experiments and ablation study that we conducted validate the effectiveness of the proposed model. Additionally, we released a new dataset to encourage the community to further explore the multi-document representation learning. These contributions advance research in scientific literature mining and provide a useful resource for the community to further investigate this new but important topic.

## Limitations

Despite its empirical success, the use of SMRC$^2$ has two notable limitations. Firstly, its application is restricted as it requires interconnections between documents (e.g. a citation graph) to form the multi-document graph. Secondly, the method incurs high computational costs due to its inclusion of a Language Model, two Graph Neural Network-based modules, and multiple operations to be trained in tandem. Furthermore, this multi-MLP embedding concatenation process makes it challenging to parse the encoded embeddings, thus potentially impacting performance. These limitations suggest that further investigations are needed to address the scalability and generalizability of the proposed model to improve its applicability in a broader range of scientific data.

## Ethics Statement

After carefully reviewing the ACL Ethics Policy, we are committed to show our respect and obey to consent all.

## Acknowledgements

We gratefully acknowledge support from NSF, Award # 2122232 - SCISIPBIO: Constructing Heterogeneous Scholarly Graphs to Examine Social Capital DuringMentored K Awardees Transition to Research Independence: Explicating a Matthew Mechanism.

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

## Training Settings

The parameter settings of trainable module are detailed in Table 7 and Table 8.

| GCN | | Training | |
|---|---|---|---|
| **Parameter** | **Value** | **Parameter** | **Value** |
| Number of layers | 1, **2**, 3 | learning rate | $10^{-3}$, $10^{-5}$ |
| emb size | 100 | dropout | 0.3, **0.6** |
| hidden size | 512 | batch_size | **20**, 50 |
| | | weight decay | $10^{-3}$ |

Table 7: The experimental settings of graph and training with the best parameter settings are highlighted.

| Pre-trained Model | | Transformer | |
|---|---|---|---|
| **Parameter** | **Value** | **Parameter** | **Value** |
| emb size | 768 | Num of Attention Head | 8, **16**, 32 |
| max length | 256, **512** | Num of Layers | 16, 32, **64**, 128 |
| | | dimension | **512**, 768 |

Table 8: The experimental settings of language model with the best parameter settings are highlighted.

## Dataset Statistics

**Category** *Bio-Sci* contains 11 different paper categories in total: Cardiovascular diseases, Kidney disease, Respiratory diseases, Diabetes mellitus, Digestive diseases, Hiv, Hepatitis, Mental disorders, Musculoskeletal disorders, Neoplasms, Neurological disorders, of which "Cardiovascular diseases" and "Mental disorders" take the most percent 26%, 18% respectively. Others share evenly for the rest.

| Dataset | Nodes | Edges | Avg. Node Degree |
|---|---|---|---|
| Ogbn-Arxiv | 169,343 | 1,166,243 | 13.7 |
| Ogbl-Citation2 | 2,927,963 | 30,561,187 | - |
| Bio-Sci | 32,330 | 222,652 | 13.7 |

Table 9: Statistics of datasts

## SciSpacy Details

For SciSpacy usage, we use them to obtain the entities with "en_core_sci_sm" and "en_ner_bionlp13cg_md" corpus because these two settings are trained in large-scale biomedical domain corpus. The f1 score is 67.87% on the entity extraction task for "en_core_sci_sm", and 76.75% for "en_ner_bionlp13cg_md". We mainly leverage "en_ner_bionlp13cg_md" in our experiments. What we do for linking entities is we first extract

the entities and retrieve the BKG to see if there are matches for them.

## Message Passing Mechanism Details

What we do for using MPM is that we use the semantic information, $h_i^E$ and $h_i^T$, learned from Section 3.1 to initialize $h_i$. As we obtain the initialized document embedding, the MSG function is for finding and collecting the embeddings of nearby nodes for node i and the AGG function will be used to perform aggregation (e.g., concatenation, pooling etc.) of embeddings in terms of node i and its neighbors. Eq. 8 will update the representation of node i iteratively and thus both the semantic information of node i itself and its neighbors can be learned and will be used in the following training procedure.