# OpenReview forum: "Content- and Topology-Aware Representation Learning for Scientific Multi-Literature"
_EMNLP/2023/Conference — EMNLP 2023 Main_

### Official Review · Reviewer_FNgn · 2023-07-31

**Typos Grammar Style And Presentation Improvements:** 1. The title of the paper is misleadi…
**Soundness:** 3

**Excitement:**

3: Ambivalent: It has merits (e.g., it reports state-of-the-art results, the idea is nice), but there are key weaknesses (e.g., it describes incremental work), and it can significantly benefit from another round of revision. However, I won't object to accepting it if my co-reviewers champion it.

**Paper Topic And Main Contributions:**

The paper is on learning better representations for scientific papers. Specifically, the authors propose the SMRC^2 model that has two main components: 1) semantic information learning phase, where the input documents' salient features are captured using a GNN encoder and LM encoder and jointly trained with Wasserstein distance constraint; 2) relatedness information learning phase, where the input documents are represented using the vectors learned in phase 1 and a multi-document citation graph is created. A GNN decoder predicts the masked edges in the citation graph. Such a pipeline aims to capture the content, and topological information originating from the citation links, to better represent scientific literature.

**Questions For The Authors:**

Question A: Is the goal to model multiple documents only in the biomedical domain? If not, why not compare against well-known scientific benchmarks like S2ORC[1]?

Question B: What are the documents labels (mu(y)), as mentioned in line 327, and how are they obtained?

Question C: What is the motivation behind creating a new dataset? How are the 10 papers in the paper group in the Bio-Sci dataset selected? How big is the citation graph, and how is diversity maintained? Details around dataset creation, methodology, correctness, comparison with existing benchmarks, and thorough analysis of citation relations are necessary for understanding the main contributions of the paper.

Question D: As mentioned in line 398, why is the VGAE trained in a self-supervised manner, and how does the approach exactly differ from Hou et al. 2022's method?

Question E: What is the dimension of the representation learned for each paper in the group? How big is the unified representation (H_m, as mentioned in line 423)? Did concatenation of h_i's give the best representation? Detailed analysis of the learned unified representation is required to understand the effectiveness of the "related information learning" module.

Question F: The baselines chosen on the Multi-XScience dataset seem old. Why have recent models like KGSum[2] not been chosen?

Question G: In section 4.5, how is the compactness of the t-SNE clusters determined? There are no numbers to support the claim.

Question H: How did the SMRC model perform with KL divergence? Giving explicit numbers to showcase the effectiveness of Wasserstein distance would be very useful.

-------

[1] Lo, Kyle, et al. "S2ORC: The Semantic Scholar Open Research Corpus." Annual Meeting of the Association for Computational Linguistics (2020).

[2] Wang, Pancheng, Shasha Li, Kunyuan Pang, Liangliang He, Dong Li, Jintao Tang and Ting Wang. "Multi-Document Scientific Summarization from a Knowledge Graph-Centric View." International Conference on Computational Linguistics (2022).

**Reasons To Accept:**

The SMRC^2 model incorporating content and topological structure is exciting and makes intuitive sense. Using GNNs and LMs to encode input documents shows higher performance on the chosen benchmark. Moreover, applying Wasserstein distance as a constraint to model the output distribution has proven effective in various graph learning theories but has yet to gain much attention in the NLP community. It would be worthwhile to study the application of Wasserstein distance as opposed to KL divergence for NLP tasks.

**Reasons To Reject:**

The experiments and analysis of the proposed SMRC^2 are weak. Clear reasoning for the chosen baselines and constructed dataset is missing. The paper was hard to follow, and the central claims of the paper need experimental justification.

**Reproducibility:**

3: Could reproduce the results with some difficulty. The settings of parameters are underspecified or subjectively determined; the training/evaluation data are not widely available.

**Reviewer Confidence:**

4: Quite sure. I tried to check the important points carefully. It's unlikely, though conceivable, that I missed something that should affect my ratings.

---

> ### Author Rebuttal · Authors · 2023-08-28
>
> We sincerely thank you for taking the time to review our work and recognize the value of the insightful comments you have made. We are committed to addressing the issues you have noted and endeavoring to assure the quality of this study.
>
> **Q A/C:** Is the goal only in biomedical domain, why not use another dataset such as S2ORC? What's the motivation of creating new datasets? (Statistics Details)
>
> **A:** The proposed method is adaptive to different domains. First, our method is extensible and not depend on any domain-specific graph and document. In the Section 4.1, we have evaluated our method on Ogbn-Arxiv, Ogbn-Citation2 and Multi-XScience datasets which are from the computer science domain. Second, we introduce our method based on Bio-Sci corpus for better understandings because high-quality graphs are easy to construct from public biomedical knowledge graphs.
>
> We admit that S2ORC is a high-quality and popular dataset, however, the S2ORC contains 81.1M papers and requires incredible computing and storing resources consumption. Because of limited conditions, we choose Obgn benchmarks and Multi-XScience benchmarks, which have the medium scale and is more suitable for method validation. It should be emphasized that the datasets we used have proved the effectiveness of the method and can be extended to massive data sets.
>
> We also made task and statistic comparisons with these benchmarks as shown in Table 1 and Table 8, respectively. Our motivation for creating Bio-Sci is that we want to include both content and topology information into learned representation which can be used not only for single-document tasks but also multi-document tasks. To achieve this goal, the dataset should be able to contain both text, citation graph and summary information. Thus, we attempt to create a new dataset which can be easily accessed and used. Our Bio-Sci is a continuous work of our whole project which motivates us to further create this dataset for the community who have interests in BioNLP as well as Scientific NLP. We highly appreciate your suggestion for using S2ORC. We indeed considered utilizing this high-quality dataset, however, the overall complexity for obtaining and processing this dataset can be too high for us since it contains 81.1M papers and we need to construct a citation graph for the whole collection. These discouraged us for giving up S2ORC and turned to a replacement of which Obgn benchmarks contains both text and citation information, Multi-XScience benchmarks provide summary information. We highly appreciate your suggestions and will try our best for exploring the usage of S2ORC dataset!
>
> **Q B:** Document labels description problem
>
> **A:** As indicated in line 488, the document labels are their MeSH terms. The Mesh Terms are the category entries defined by PubMed. Specifically, each paper in PubMed is associated with its Mesh Term, we use off-the-self tool called pubmed_parser (https://github.com/titipata/pubmed_parser) to obtain the mesh term and put the terms as the document labels.
>
> **Q D:** Why VGAE is trained in a self-supervised manner, different from Hou et al. 2022?
>
> **A:** As indicated in line 385, our assumption here is that the citation relation between two papers can be inferred by examining their respective contents, corresponding with the standard approach in GNN for inferring edge connections based on node features. We have the citation graph of given paper group and ideally the topology information can be learned through enabling VGAE to learn the invisible citation relation (citation masking). To further reduce the potential noise, we also use pair-wise node cross-relation representation in each layer instead of last-layer hidden representation to enable sure the shared pattern between node i and its neighbors can be learned (please refer to the answer to Q B for reviewer LfTX). As indicated in line 395-398, the main difference between our methodology and Hou et al. 2022 is that they mask a subset of node features and the corresponding hidden representation and train their model in an unsupervised setting which back for great performance of node classification task as can be seen in the Section 3.2 of its original paper. However, our node features are ensured by the joint text- and entity-level representation learned under Wasserstein Distance constraint and thus we resort to masking the edges to enable the topology information learning. By different masking strategies and pair-wise cross relation representation usage, consequently the loss function and the learning goal for decoder is also different. We will provide additional narrative to make this content more noticeable.
>
> **Q E:** Dimension of the representation learned? Is concatenation the best configuration?
>
> **A:** As indicated in the Table 6, the dimension of the representation for each paper is 512. The unified representation can be a matrix of 10x512. We did try attention mechanism for combinations of $h_i$, however, the results for multi-doc task (i.e., multi-doc summarization) decreased to 20.01 which is not comparable with 26.03 of using concatenation. We thought that is because the attention multiplication can introduce unnecessary/irrelevant noises to the shared patterns learned by the pair-wise cross-correlation operation (Please refer to the answer to **Q A** for reviewer LfTX). We further replace the pair-wise cross-correlation with last layer's hidden representation with attention combination, the result for multi-doc summarization is 25.31 which is close to 26.03 and further confirm our assumption that the cross-correlation computation plays similar/better role than attention mechanism. We also tried using mean pooling for combinations of $h_i$ and the result of 22.51 is also not as good as the settings reported in our paper. We may can't conclude that with the assistance of cross-correlation computation, concatenation is the best for forming unified representations. But it is the most computation-effective way to obtain the best performance so far. Due to the time limitation, we only test the attention mechanism and mean pooling methods, however, we will further try other methods for evaluate the effectiveness of the whole frame. Many thanks for your useful suggestions, we will include these results and analysis in our revision.
>
> **Q F:** Baseline is old for MDS, why not included KGSum?
>
> **A:** Following your suggestions, we compared the proposed method with KGSum on Multi-XScience and Bio-Sci. As following, the experimental results on two public datasets show that our method performs outperforms KGSum. We further check their method and attribute these results to the fact that our method leverages a Wasserstein Constraint for modeling the interactions between entity-relation graph and semantic text information entity while KGSum only learns from salient entity-sentence graph.
> | Model |  | Multi-XScience |  |  | Bio-Sci |  |
> | :---: | :---: | :---: | :---: | :---: | :---: | :---: |
> |  | ROUGE-1 | ROUGE-2 | ROUGE-L | ROUGE-1 | ROUGE-2 | ROUGE-L |
> | KGSum | 30.16 | 6.24 | 25.37 | 34.98 | 6.91 | 30.07 |
> | SMRC2 | 31.47 | 6.73 | 26.03 | 36.23 | 7.31 | 30.63 |
>
> **Q G:** How the t-SNE's parameters are chosen?
>
> **A:** The parameters are set experimentally. See the details in https://projector.tensorflow.org/, I have uploaded the learned representation of 100 samples to this tool and tuned the parameters to obtain the best results as shown in Section 4.5. The figure is produced by matplotpib library using the parameters we obtained from the above link.
>
> **Q H:** How the KL/JS is used for SMRC?
> **A:** As shown in line 338 and Eq. 6, once we have the results of $y^T$ and $y^E$, the KL can be detailed as $L(y^T, y^E) = y^T*log(y^T/y^E)$, the ordering can be changed and for JS computation we plus the opposite order together[2]. We highly appreciate your kind suggestions and we did include the analysis to showcase the effectiveness of Wasserstein distance as can be seen in Table 5 and Section 4.4. We evaluate the different distribution alignment methods on the three tasks and among them, model equipped with Wasserstein distance performs the best.
>
> **In Responding to the “Reasons To Reject”:**
>
> **1.** Writing problem
>
> **A:** Many thanks for your suggestions towards our writing issues, we will follow your suggestions and improve our writing in the revision which will include all the suggestions and additional experiments as well as the analysis. Thanks again.
>
> **Reference**
>
> [1] Y Lu, Y Dong, L Charlin. Multi-XScience: A Large-scale Dataset for Extreme Multi-document Summarization of Scientific Articles. Proceedings of the 2020 on Empirical Methods in Natural Language Processing.
>
> [2] https://pytorch.org/docs/stable/generated/torch.nn.KLDivLoss.html

---

### Official Review · Reviewer_phqf · 2023-08-05

**Soundness:** 4

**Excitement:**

4: Strong: This paper deepens the understanding of some phenomenon or lowers the barriers to an existing research direction.

**Paper Topic And Main Contributions:**

To represent multi-document texts in different levels, this paper proposes a new document representation learning method, which contains sementic representation with entity relations for each document and constructs a multi-doc graph based on relatedness information in document citation networks.

**Reasons To Accept:**

Beside text embedding using language model, this paper introduces entity-relation graph representation, and relatedness representation to form the multi-doc graph. In text presentation, Wasserstein Distance is introduced to minimize the distance between entity-level and text-level representation. The loss contains loss functions for both document-level semantic representation learning and relatedness representation. The results on three different tasks shows the efficiency of this model.

**Reasons To Reject:**

The scope of this model is limited, since it requires both document types and citation networks among documents.

**Reproducibility:**

4: Could mostly reproduce the results, but there may be some variation because of sample variance or minor variations in their interpretation of the protocol or method.

**Reviewer Confidence:**

2: Willing to defend my evaluation, but it is fairly likely that I missed some details, didn't understand some central points, or can't be sure about the novelty of the work.

---

> ### Author Rebuttal · Authors · 2023-08-28
>
> Many thanks for your suggestions, we highly appreciate your time and efforts spent on reviewing our paper. To address your concerns for our work, we will further explore the potential for reducing the computation requirement and flexibility in the continuous work. For example, document hyperlinks can also be used to model the multi-doc level graph and the pattern can be different from citation relations. Also, cascade training schema and the power of large language models can be used to boost our model. Thanks again!

---

### Official Review · Reviewer_LfTX · 2023-08-11

**Soundness:** 3

**Excitement:**

4: Strong: This paper deepens the understanding of some phenomenon or lowers the barriers to an existing research direction.

**Paper Topic And Main Contributions:**

This paper is about representation learning in scientific literature at the multi-document level. The authors propose a novel approach called SMRC$^2$ (Scientific Multi-Literature Representation Learning with Content- and Topology-Aware Representation Learning). They propose an approach that combines both semantic content and topological relatedness information.

The main contributions of this paper are as follows:
- Introducing SMRC$^2$, which effectively combines semantic content and topological relatedness to create a superior learning space for scientific literature representation learning.
- The authors introduce Wasserstein distance to address the imbalance learning problem during the semantic learning stage.
- Experiments conducted by the authors demonstrate that the SMRC$^2$ method achieves excellent results.
- The authors release a new dataset to encourage further exploration of multi-document representation learning in scientific literature mining.

**Questions For The Authors:**

A: Equation **(9)** proposed by the authors in *Section 3.2* seems to compute the cross-correlation of pairs of nodes at the same level and then concatenate them together because *directly applying the final-layer hidden representation can be difficult for producing the reconstruct matrix*. However, it seems that applying the last level of hidden representation directly is also feasible in this context, and the approach of Eq. **(9)** introduces more unnecessary complexity, can the authors elaborate on the reasons for doing so?

B: How exactly do the GNN used by the authors in Section 3.2 perform message passing?

C: Have the authors conducted experiments to discuss the effectiveness of the masking strategies? What are the specific results?

**Reasons To Accept:**

- Representation learning for multi-scientific documents is a research direction of interest in the NLP community, and this work may be of interest to many related researchers.

- This paper is easy to follow. In describing the methods they use, the authors have analysed their reasons for adopting them and compared the advantages and disadvantages of other methods. In addition, the authors provide a detailed and comprehensive analysis of their experimental results, making it easy for the reader to understand the conclusions reached by the authors.

- The authors introduce the *Wasserstein Constraint* for multi-view semantic-level representation learning, which is experimentally demonstrated to have considerable improvement over KL or JS divergence. This discovery is a noteworthy contibution. It is possibly a worthwhile practice for scientific document representation learning after passing further validation.

- The authors provide a new Bio-Sci dataset of high quality in this paper, which is valuable for the evaluation of research in this field.

**Reasons To Reject:**

- Some of the detailed information in this paper is not provided in its entirety as the authors describe their methods. For example, when using GNN for relatedness information learning in *Section 3.2*, how exactly is the message passing mechanism of GNN designed?

- The experiments conducted in this paper seem to me to be insufficient to some extent. Firstly, the experiments carried out by the authors compare few baseline methods and do not have more recent work from recent times, more experimental methods should have been included for comparison. Also, more ablation experiments should have been conducted to demonstrate the effectiveness of some of the strategies proposed by the authors (e.g., the masking strategy).

- The authors did not show and analyse the statistics of the datasets used in the article. What are the specific domains of these datasets? How much literature and citations are included? These are important to analyse the scope of applicability of the methodology of this article and to analyse the reliability of the experimental results.

**Reproducibility:**

4: Could mostly reproduce the results, but there may be some variation because of sample variance or minor variations in their interpretation of the protocol or method.

**Reviewer Confidence:**

4: Quite sure. I tried to check the important points carefully. It's unlikely, though conceivable, that I missed something that should affect my ratings.

---

> ### Author Rebuttal · Authors · 2023-08-28
>
> We sincerely appreciate your useful suggestions. We notice there exists some implicit and unclear method descriptions which lead to your confusions and concerns. We admit to enhance the writings more explicit and some additional empirical results are provided to further support my response and hopefully can reduce some of your concerns.
>
> **Q A**: Why Equation 9 is applied which is more complex instead of using the last layer hidden representation?
>
> **A**: This is a commonly-used technique in representation learning inspired from self-supervised learning on graph [1]. Our validation confirms that a combined-layer representation outperforms the last-layer representation due to the latter's tendency to forget long-distance neighbors with an increasing number of layers.
>
> Further, we indeed tried to use the last layer of hidden representation initially. However, the performance of the citation adjacent matrix reconstruction was suboptimal. Upon investigations, we attributed this to the fact that the embedding of l-th layer can contain irrelevant information from neighbors which leads to potential noises inclusion. Subsequently, we adopted the strategy of computing the cross-correlation of pairs of nodes to identify the shared patterns between node i and its neighbors and thus alleviate the long-distance and irrelevant inclusion problem. We apologize for any confusions caused in our paper and we promise to provide more details in the revision.
>
> **Q B**: How the message passing mechanism of GNN is performed?
>
> **A**: Thanks for this question. Typically, the message passing mechanism (MPM) follows an iterative schema of updating the node representation based on the aggregation from its neighbor nodes[2]. The powerful MPM enables GNN to realize the topological information. However, due to the page length limitation, we resort to saving the space for other details of our overall frame instead of these common-used MPM information. Our primary design goal in this work was to enable the nodes within the multi-doc graph to be aware of the information of their neighbors through the use of MPM. To this end, what we do is that we use the semantic information learned from Section 3.2 to initialize $h_{i} (h_{i}^{0} = [h_{i}^{E}, h_{i}^{T}])$ as can be seen in line 377. As we obtain the initialized embedding, the MSG function is for finding and collecting the embeddings of nearby nodes for node i and the AGG function will be used to perform aggregation (e.g., concatenation, pooling etc.) of embeddings in terms of node i and its neighbors. Eq. 8 will update the representation of node i iteratively and thus both the semantic information of node i itself and its neighbors can be learned and will be used in the following training procedure. Due to the page length limitation, we are not able to put these details. However, we recognize that there is an opportunity to further enhance our writing to reduce any confusion, and we will address this problem in the revision.
>
> **Q C**: The experiments on evaluating the effectiveness of masking strategy.
>
> **A**: Thank you for your constructive feedback regarding our approach. We did try different masking ratio (this parameter can be found in line 400) and recognize that the selection for masking ratio is an experimental selection. We have evaluated different masking ratio strategies. Specifically, we set the ratio as 10%, 20% and 30% and the results are detailed as below:
> |Settings | Document Classification | Citation Prediction | Summarization|
> |:---:|:---:|:---:|:---:|
> |10% | $86.19\pm 0.11$ | $74.13\pm 0.31$ | 24.91
> |20% | $88.41\pm 0.21$ | $73.97\pm 0.19$ | 26.03 |
> |30% | $84.17\pm 0.07$ | $67.89\pm 0.13$ | 22.51|
>
> What we can conclude from the results is that masking ratio can influence the effectiveness of overall performance since the masking strategy is associated with the relatedness information learning. When masking more edges, the performance of citation prediction will inevitably decrease since the complexity of recovering can be higher. However, further experimentation revealed that 5% masking for citation prediction produced a result of $70.21\pm 0.11$, suggesting a decrease due to the risk of overfitting.  We can also notice that the performance of other tasks is influenced since there can be semantic information lost and noise introduction due to the update of hidden representations which further validates the necessity of Eq. 9. We will enhance the analysis writing and include this part in the revision.
>
> **In responding to the “Reasons To Reject”:**
>
> **1.** GNN and Message Passing Mechanism Design:
>
> **A:** Please refer to the answer for **Q B**.
>
> **2.** Insufficient Experiments:
>
> **A:** We appreciate your constructive feedback regarding the experiments. To address your concern, we further conducted supplementary experiments beyond the basslines chosen in our paper, the results can be seen as follows:
> | Model |  | Multi-XScience |  |  | Bio-Sci |  |
> | :---: | :---: | :---: | :---: | :---: | :---: | :---: |
> |  | ROUGE-1 | ROUGE-2 | ROUGE-L | ROUGE-1 | ROUGE-2 | ROUGE-L|
> | KGSum | 30.16 | 6.24 | 25.37 | 34.98 | 6.91 | 30.07 |
> | $SMRC^{2}$ | **31.47** | **6.73** | **26.03** | **36.23** | **7.31** | **30.63** |
>
> We follow the suggestions of **Q F** from Reviewer FNgn. KGSum is a KG-enhanced method for Multi-Document Summarization task. Its original paper conducted experiments on Multi-XScience (we run it again on the Multi-XScience in terms of fair comparisons), we further run the code on our Bio-Sci. As can be seen above, it is evident that our method surpasses the other baselines which we attribute to the fact that our method leverages a Wasserstein Constraint for modeling the interactions between entity-relation graph and semantic text information entity while KGSum learns only from salient entity-sentence graph. We will host additional narrative to enhance the experiment section.
>
> **3.** No dataset details.
>
> **A:** We have included the statistics in the Table 8 in Appendix and elaborate them in data section. The number for nodes, edges and avg. node degree are 32,330, 222,652 and 13.7, respectively. The comparisons against two benchmark datasets are displayed in the Appendix and the meta-data. We also detail construction methods in Section 4.1. Because of space limitations, we cannot put these details in the main text, but we will add more data information to the appendix in the revised version and more detailed data description will be available in the project web page.
>
> **Reference:**
>
> [1] Qiaoyu Tan et al., MGAE: Masked Autoencoders for Self-Supervised Learning on Graphs.
> [2] Hamilton, William L., Graph Representation Learning. Journal of Synthesis Lectures on Artificial Intelligence and Machine Learning.

---

### Meta-Review · Area_Chair_HPKk · 2023-09-15

**Recommendation:** 5

**Metareview:**

This paper introduces SMRC^2, a new method for representation learning for scientific literature. The paper also introduces a new bio-sci dataset for evaluating systems in this field. SMRC^2 is demonstrated to be an effective strategy, useful for downstream tasks.

SMRC^2 incorporates a *Wasserstein constraint*, which the authors empirically demonstrate to perform considerably better than alternatives; this is likely to have implications beyond SMRC^2 itself.

The paper initially lacked details, including specifications of some model components as well as comparisons to recent baselines; the authors have been diligent in addressing these in their responses, which when incorporated into the main text will much improve the paper. The research presented in this paper is of high quality, but initially held back by a lack of clarity. Two reviewers raised their soundness scores in response to the rebuttals, mainly because additional details around implementation and baselines were provided.

---

### Decision · Program_Chairs · 2023-10-07

**Decision:**

Accept-Main

**Comment:**

This paper introduces SMRC^2, a new method for representation learning for scientific literature. The paper also introduces a new bio-sci dataset for evaluating systems in this field. SMRC^2 is demonstrated to be an effective strategy, useful for downstream tasks.

SMRC^2 incorporates a *Wasserstein constraint*, which the authors empirically demonstrate to perform considerably better than alternatives; this is likely to have implications beyond SMRC^2 itself.

The paper initially lacked details, including specifications of some model components as well as comparisons to recent baselines; the authors have been diligent in addressing these in their responses, which when incorporated into the main text will much improve the paper. The research presented in this paper is of high quality, but initially held back by a lack of clarity. Two reviewers raised their soundness scores in response to the rebuttals, mainly because additional details around implementation and baselines were provided.